# Psychosocial Consequences of Spinal Cord Injury: A Narrative Review

**DOI:** 10.3390/jpm12071178

**Published:** 2022-07-20

**Authors:** Maggi A. Budd, David R. Gater, Isabella Channell

**Affiliations:** 1Spinal Cord Injury/Disease Service, VA Boston Healthcare System, Boston, MA 02130, USA; bellemc@bu.edu; 2Department of Psychiatry, Harvard Medical School, Boston, MA 02115, USA; 3Center for Bioethics, Harvard Medical School, Boston, MA 02115, USA; 4Department of Physical Medicine and Rehabilitation, University of Miami Miller School of Medicine, Miami, FL 33136, USA; dgater@miami.edu; 5Christine E. Lynn Rehabilitation Center for the Miami Project to Cure Paralysis, Miami, FL 33136, USA; 6The Miami Project to Cure Paralysis, University of Miami Miller School of Medicine, Miami, FL 33136, USA; 7Mental Health Counseling and Behavioral Medicine Graduate Program, Boston University School of Medicine, Boston, MA 02128, USA

**Keywords:** psychosocial, spinal cord injury, wholistic, behavior, psychology, social work, rehabilitation, adaption, disability

## Abstract

Consequences of a spinal cord injury (SCI) entail much more than damage to the spinal cord. The lives of people with SCI, along with those around them, experience profound long-lasting changes in nearly every life domain. SCI is a physical (biological) injury that is inextricably combined with various psychological and social consequences. The objective of this review is to present psychosocial challenges following SCI through the biopsychosocial model, beginning with acknowledgement of the larger societal effects of ableism and stigma before addressing the many unique psychosocial aspects of living with SCI. Included in this review are qualitative studies and systematic reviews on current psychosocial outcomes and consequences. This paper attempts to structure this information by dividing it into the following sections: relationships and family; changes in finances and employment; issues related to the person’s living situation; community reintegration; factors associated with mood and coping (e.g., depression, anxiety, substance use, and PTSD); self-harm behaviors (ranging from nonadherence to suicide); effects of traumatic brain injury; considerations regarding sexual health; aging with SCI; and concludes with a brief discussion about post-traumatic growth. Cultivating an understanding of the unique and interrelated psychosocial consequences of people living with SCI may help mitigate the psychosocial aftermath and serve as a reminder to providers to maintain a person-centered approach to care.

## 1. Introduction

Acquiring a spinal cord injury (SCI) involves more than damage to the spinal cord; the lives of people with SCI along with those around them experience profound long-lasting changes [1,2]. Depending upon the level and completeness of SCI, medical comorbidities besides paralysis may likely include neurogenic bradycardia, neurogenic orthostatic hypotension, circulatory hypokinesis, adaptive cardiomyopathy, neurogenic restrictive lung disease, neurogenic obstructive lung disease, obstructive sleep apnea, neuropathic pain, spasticity, reflex neurogenic bladder, reflex neurogenic bowel, neurogenic erectile dysfunction, neurogenic infertility, neurogenic skin, neurogenic obesity, heterotopic ossification, osteopenia/osteoporosis, and the metabolic syndrome including diabetes mellitus, dyslipidemia and hypertension [3]. Of note, the International Classification of Functioning, Disability and Health (ICF) has been developed to provide a universally accepted framework to classify and describe function, disability and health involving body functions, body structures, activities and participation, and environmental factors for any specific diagnosis. When applied to chronic SCI, an ICF Comprehensive Core Set included 44 categories from body function, 19 from body structures, 64 from activities and participation, and 41 from environmental factors [4]. As many of these comorbidities will be reviewed as separate manuscripts in this Special Issue, we will focus our attention instead on the interaction between biological, social, and psychological influences on the person with SCI. The sheer number of medical comorbidities certainly influences mental health outcomes in this population [5,6,7,8], as is the case in non-SCI individuals with greater numbers of medical comorbidities [9,10].

In 1977, George Engel formulated a philosophical and practical approach to clinical care called the “biopsychosocial” model [11]. Philosophically, this model helps others appreciate a person’s experience of medical issues by considering the dynamic interactions of physiological factors (bio) with the psychological and personal (psycho) and societal components (social). From a practical standpoint, the biopsychosocial model expands the standard medical model enabling multiple disciplines to approach the multifaceted components of the “person” and “social environment” [12] that directly affect subjective wellbeing and overall outcomes in the context of medical problems. For example, problems with mood, relationships, or personal sense of meaning can result in maladaptive behaviors [13] (e.g., alcohol misuse) that can cascade into further physical, social, and functional impairments, compounding challenges in an already difficult circumstance. Figure 1 depicts psychosocial factors that can be relevant after one acquires an SCI.

Although we cannot fully separate bio–psycho–social components, this paper will attempt to parse out and articulate psychosocial consequences that are common for people after SCI, excluding biological factors that will be covered elsewhere in this Special Edition. Therefore, we aimed to review the literature that describes current psychosocial challenges within the SCI population, as well as provide considerations that promote quality of life from a biopsychosocial perspective. This review focused on two key concepts (spinal cord injury and psychosocial consequences) along with associated key terms and was conducted using PubMed and Knowledge Library at VISN 1 (va.gov) (accessed on 12 February 2022 and 19 February 2022). Each section presented will have some overlap, yet each will have a presence of its own. In this paper, non-traumatic and traumatic SCI will not be differentiated, as the lived psychosocial experiences are likely similar. Childhood SCI comprises a small minority of SCI and involves unique psychosocial issues. Therefore, for reasons of time and space this review will only focus on adults with an acquired SCI from unspecific etiologies. This review begins by acknowledging the larger societal effects of ableism and stigma [14] followed by presenting several features unique to the psychosocial aspects of living with SCI.

## 2. Appreciating the Ableist World in Which People with SCI Live

The larger social–psychological consequences affect all aspects of a person’s life following an SCI [1,14]. Surprisingly, little has changed regarding the social influences on the experience for people with SCI. Social psychologists note that the experiences of social judgment, affect, and behavior for both the observer and the person observed (with SCI) are similar to experiences had 60 years ago [15].

Ableism is disability-based discrimination that results from conceptualizing able-bodied people as “normal” and superior to people with disabilities. Despite its presence, stigma and bias towards people with disabilities are studied much less frequently than other attitude-relevant domains such as age, race, or gender. Although bias can influence judgment and actions in either a positive or negative direction [15,16], studies revealed a general negativity towards people with disabilities [14,17,18]—a negative attitude that people with disabilities expressed themselves [17].

Misconceptions about the lived experiences for people with SCI are regrettably common and can inadvertently influence an emotional bias (e.g., prejudice), cognitive bias (e.g., stereotype), or behavioral bias (e.g., discrimination). These can result in self-stigma, public-stigma, or professional or institutional stigma. Self-stigma is when a person sees themself in a stigmatized way. According to research, self-stigma appears to change over the course of injury [19]. Self-stigma is most prevalent during the first two to three years of SCI [20], although the experience of stigma often persists over the course of one’s lifetime [21]. 

People with high self-stigma after SCI are highly conscientious of how they appear and relate to others, contributing to a sense of social detachment. Descriptively, these individuals were inclined to be younger, single, to spend more time in rehabilitation [19], and use power wheelchairs [19,22]. Regarding social connections, higher self-stigma was found to influence a person’s social choices. Whether disassociating from old friends or connecting with a new set of peers, stigma was associated with a sense of social disconnection [23].

For people without SCI, a major implicit bias with SCI is the “spread effect” [20,24] or “ineffectual bias” [16] that assumes an individual’s disability also negatively affects the whole person such as their intelligence, abilities, or personality. These biases can result in more paternalistic attitudes and less focus on the person’s agency. Individuals with SCI report experiencing this through beliefs that others often felt pity for them, that people typically felt uncomfortable around [23] and that people generally had lower expectations of them because of their injury [22].

Stigmatizing attitudes from people without disabilities can stem from a curiosity or simple ignorance about living with SCI [15,24]. Not knowing the facts that most people adapt well to SCI, some people may project onto the person with SCI their own perceptions of catastrophe they would expect themselves to have if they were disabled, or they might imagine the person with SCI is overly fragile. These catastrophe and fragility biases can also extend into how providers respond to patients with SCI. Providers with these biases can give more severe diagnoses and have overall diminished expectations from the person with SCI [16].

A potentially harmful implicit bias is when people ascribe a person’s SCI with how the SCI occurred. Academics refer to the moral model of disability [14] or the *just-world hypothesis* to explain negative reactions nondisabled people have toward people with disabilities [15,24]. People unconsciously believe good things happen to good people and bad things happen to bad people, or that morality and justice are matched. Therefore, when bad things happen such as SCI, the affected person may have done something to curate the bad thing. Studies related to stigma and SCI that appear to support these hypotheses through findings where stigma was developed based on how a person acquired their SCI (if it was controllable like in an automobile accident) or uncontrollable (such as being hit by an impaired driver), or stigma being based on the amount of assistance a person needs from others [22], or stigma being centered on the person’s level of physical activity [18].

Feeling any form of stigma can affect psychosocial outcomes such as social disconnection [23], a sense of injustice [12], greater depressive symptoms, reduced self-efficacy, and decreased quality of life [19]. Living with SCI is complicated, and stigma and bias are only one of several enduring psychosocial consequences a person experiences following SCI [21].

Generating a constructive view of living with disability is most helpful to reduce stigma and ableism. Highlighting the apparent struggles and dependencies resulting from a disability can be aversive. Although some challenges are genuine, noting them does not add a constructive view of living with disabilities. Over-acknowledging successes or challenges perpetuates negative attitudes toward people living with disabilities. Rather than adding constructive views of living with disability, these attitudes often engender pity or fear.

The purpose of this review is to present the literature identifying the most updated psychosocial information regarding consequences after SCI. When available, we shared the most recent guidelines for navigating psychosocial consequences. Included are studies examining psychosocial outcomes and complications, and systematic reviews to provide the most current information and guidance. Additionally, included are qualitative studies to present the patient perspective. The paper is divided into sections presenting the far-reaching effects on relationships and family [25]; changes in finances and employment [6,26]; issues related to the person’s living situation; the literature on community reintegration; factors associated with mood and coping; SCI and traumatic brain injury; aging with SCI; and, concluding with positive outcomes such as post-traumatic growth.

## 3. Psychosocial Consequences

Early models of disability posited that disability was a static experience, and concerns waned after one “adapted to disability” [2]. Undeniably, the majority of psychosocial consequences are most intense immediately after SCI [20,22,27], and they qualitatively differ between immediate and long-term effects of SCI; however, many psychosocial consequences remain consistent throughout one’s life. A static view of living with disability neglects the reality of dynamic and continual psychosocial challenges over time [28]. SCI is a permanent condition requiring lifelong, daily adaptations for both the person with SCI and those caring for them [25,29].

## 4. Relationships and Role Changes

Living with SCI often forces individuals to re-evaluate and re-construct their personal and social goals and identities in their family and social systems as a result of transitioning to more dependent-functioning and changes in emotional, psychological, economic, environmental, and social stressors [12,23,25,30,31,32]. Such physical limitations of SCI may initially disrupt the original way in which spouses or family members interact or meet traditional expectations [2,12,33]. Despite these new and challenging stressors, individuals with SCI and their family members can better adapt to life with SCI by learning to accept the disability, staying solution focused [34,35], accentuate abilities and values [24], and utilize socialization and supportive communication [25,31,32].

### 4.1. Relationships Matter

Social relationships affect both a person’s mental health [15,24,31,36,37,38] and physical health [39]. The literature distinguishes between structural relationships and functional relationships. *Structural relationships* represent quantitative aspects of social relationships, such as the network size, and the number and frequency of contacts. *Functional relationships* refer to the qualitative features of social relationships, such as subjective sense of support, and satisfaction with the relationship [30,37]. Both types of relationships are associated with mental health problems [37] and mental health adaptation [12] for people with SCI. However, there is more weight given to the influence of functional relationships over structural relationships for one’s mental health, that is found to be independent from severity of SCI, demographics, and other health conditions [36].

Relationships change after SCI and have a direct effect on a person’s general mental health, experience of life stressors, and social opportunities [30]. Many describe how friendships gradually “fade away” after SCI, resulting in a sense of loss, confusion, and social disconnection [23]. Activities pre-SCI are often drastically changed post-SCI, and this places great challenges on maintaining or rebuilding relationships they once shared.

Most germane to psychosocial health for people with new and chronic SCI is less the quantity, but the strength of and feelings of support in their relationships [32,37]. Mental health benefits are found when a person feels they have strong, supportive relationships [32,37]. When present, these relationships add substantial meaning to the person’s life [12,31] and can significantly lessen feelings of grief and loss related to their injury [34]. Quality relationships have also been found to offer additional psychosocial benefits for people with SCI by buffering against mental and bodily stress [39] as well as financial stress [30,37], all of which are integral to daily coping and retaining resilience [1].

### 4.2. Role as Care Receiver

Individuals reportedly reconsider all of their relationships after SCI [12]. Changes in family relationships are especially troublesome, with feelings alternating between guilt and gratitude for family-members [12]. Living with loved ones after SCI generates a sense of meaning, but this can often shift to guilt for not being able to be the parent/spouse they want to be, or their inability to contribute to household tasks, which they feel adds great burden to their loved ones [12,32].

### 4.3. Loved One as the Care Provider

Similar to the person with SCI, loved ones also experience profound psychosocial changes, particularly if they become the person’s caregiver.

Family caregivers for people with SCI recounted negative and positive themes [25,29]. Family caregivers reported that several of the most distressful caregiving activities involved the emotional and physical challenges, receiving reliable and competent hired help, and the generalized strain on family relationships [32]. Life changes that affected the family system most were the inability to visit friends and other venues due to inaccessibility [12,23,25], discontinuation of family travel, and feeling more homebound following SCI [12]. However, positive themes balanced these negative experiences. Noted benefits from being a caregiver were meeting other people who they may not have met otherwise, feeling deeply appreciated, experiencing an enhanced feeling of family cohesiveness, and improved changes in self-awareness. Some caregivers even experience post-traumatic growth [25].

Often times, when spouses or family members become primary caregivers, these individuals or systems must consider role reallocation and flexibility as a means of adjusting to life post-injury [12,32,33]. This change in roles, responsibilities, and dependence can cause caregivers and care receivers with SCI to develop maladaptive relationship dynamics and self-identity reconstruction. Such maladaptive responses include (1) asymmetrical dependency, in which the caregiver and care receiver can lose their identities and freedoms; (2) unfair expectations set by care receivers on caregivers that increase burnout and frustration; and (3) caregiver’s protective behaviors facilitating dependency-inducing behaviors of the care receiver, especially for tasks in which the individual with SCI is capable of completing independently [32]. Regarding intimacy, the role transition from partner to caregiver has also been found to cause stress in and loss of sex and intimacy among partners in romantic relationships [12,32].

### 4.4. Caregiver as an Extension of the Patient

Caregiving for a person with SCI can pose multidimensional costs and significant burdens on the person providing care [12,32]. Noted risk factors for caregiver burnout include being female, living with the care receiver, depression, financial stress, lower educational attainment, number of hours spent caregiving, and feeling if there was no choice in being a caregiver [29].

It is important that providers caring for patients with dependency needs such as SCI recognize the vital influence of caregivers [32]. The caregiver is often the primary interface between the person with SCI and medical providers. Inquiring about caregiver health and wellness is providing good clinical care since research shows caregiver wellbeing directly affects the health and wellbeing of the patient [29].

There are several ways to support caregivers for people with SCI. During clinic visits, proactively explore problems and issues [29] such as needed education and skills training (e.g., proper transfers, use of lifts) to improve efficiency and reduce chance of injury; effective use of support technology (home monitors and webcams, medication, and appointment alarms; alert for self-care); and, resources for assistive services (respite care; medical day cares; meal services). Fatigue and lack of sleep were noted stressors for caregivers [25], therefore, using respite services or scheduling “breaks” from caregiving can help reduce burnout. Noted sources of strength to manage caregiving stressors were faith and support from others [25]. Caregiver support groups, faith groups, and education about burnout can benefit caregivers [32]. Caregivers being mindful of self-care is equally important for good outcomes for the person they are caring for.

### 4.5. Supporting the Role Changes

SCI often prevents caregivers and care receivers from ever returning to their routines in life pre-injury [12,32], thus it is exceptionally important that they create an “alternate narrative” and rebuild their lives, with a major focus on individuality and ability and values [24]. For the care receiver, this may look like taking on new household responsibilities or obtaining a new job that is able to accommodate their capabilities, both of which may not have been a part of their pre-SCI roles in the relationship [25]. For the caregiver, this may entail becoming unemployed in order to stay home and provide necessary care, learning to pay bills and do banking, or assuming new duties as the handy-person to maintain a house.

Caregivers and care receivers must be willing to openly communicate about caregiving boundaries and expectations in order to maintain a balance of mutual reliance and autonomous identity in the relationship [32]. Willingness for both parties in the relationship to be vulnerable, open in communication, and experiment with new, creative ways of living [32] are key factors for overcoming challenges with balancing intimacy, care needs and dependence, gender roles, and socialization [23]. Some caregivers suggested that “emotional coaching” during rehabilitation could help build interpersonal skills and coping, particularly during emotionally laden caregiving tasks such as bowel care or management of autonomic dysreflexia [32].

## 5. Vocation

Previous studies have shown that life satisfaction is positively correlated with employment, regardless of income, as vocational outcomes predict life satisfaction [28] and longevity [40,41]. Employment is especially impactful on the quality of life for people with SCI [31], notably to enhance self-esteem, foster positive role model experiences, promote optimism, positive coping, and increase motivation. Moreover, people with SCI who are employed seem to advance in psychological adjustment compared to those who are unemployed [6,40].

Research found that employability among people with SCI was associated with various multiple key factors such as functional independence, mental health and medical complications (e.g., “health burden”) [6], age, time passed since injury, sex, marital status and social support, and environment [26]. Higher levels of education and previous experience in managerial, professional, or office occupations were found to be strong predictors of individuals with SCI returning to work post-injury [26]. These individuals seem to have more opportunities for jobs that are less physically demanding [42]. Mental health comorbidities associated with obtaining employment after SCI were depression and substance abuse disorder [6].

There are societal barriers to gaining employment for people with SCI as well. These barriers include transportation issues, physical inaccessibility [41], lack of accommodations, and discrimination by employers [42]. Plum et al. [43] also addresses the concept of work disincentives for people with SCI. Those who receive high levels of benefits from Social Security Disability Insurance (SSDI), or Supplemental Security Income (SSI) may be less inclined to seek out employment after their injury in fear of losing such benefits [40,42].

In order to combat barriers to employment, people with SCI must learn to advocate for themselves, seek out job-specific training, engage in networking, and put in place necessary supports. Furthermore, people with SCI can better navigate the job field by knowing about their eligibility for benefits and awareness of disability-related workplace rights [42].

It appears that being able to return to the workplace is an important protective factor against a reduced quality of life, as employment increases opportunities for social engagement [6], as well as creates a sense of purpose and financial independence for the person living with SCI [44].

## 6. Finances

Catastrophic injuries can create enormous costs that include and extend beyond the use of health care services [26]. Such costs consist of initial hospitalization, acute rehabilitation, home and vehicle modifications, medical equipment, medication, supplies, and personal assistance services [26]. Additionally, lost wages, long-term secondary complications (i.e., UTIs, sepsis due to pneumonia, and pressure ulcers) also contributes to higher financial costs [44]. Furthermore, such accumulated costs are also affected by initial surgical intervention and the level of injury, for individuals with a higher level of injury, have an increased level of financial costs [26,44].

Another source of financial strain for persons living with SCI comes from issues with employment [26]. Many individuals with SCI have a strong desire to return to work after adapting post-injury, but unfortunately factors such as inaccessibility or transportation barriers may cause impassible challenges for individuals to return to work or find new employment opportunities [41]. Along with income lost from the person with SCI, caregivers and family members of the individual with SCI may not be able to return to their previous employment [25,27]. Consequently, this level of financial stress poses as a risk factor for caregiver burnout [29], and reduced participation and experience of loneliness for the person with SCI [30].

Mild to severe financial difficulties are associated with increased health problems, functional abilities, and overall quality of life for people with SCI [30,39]. Issues with financial difficulties have been found to contribute to feelings of unhappiness, even more so than the disability itself [30]. If feelings of unhappiness related to finances are prolonged, mental health problems may be more likely to be exacerbated [44]. However, having strong social relationships can reduce the risk of mental health issues and act as a protective factor against other adverse effects of financial strain [30,37].

## 7. Living Situation and Community Integration

Persons with SCI and their family may be at a loss after discharge from acute rehabilitation facilities and services. Rehabilitation facilities are specifically equipped and adapted for SCI-independent mobility and functioning, whereas the outside world is not. Therefore, persons with SCI and their family face more obstacles in learning to navigate the “real world” environment and society as they leave behind the familiarity and safety of their rehabilitation facility and supportive staff. This may result in leaving the individual with SCI feeling that they must cope with these challenges and adjust on their own [12], and often without sufficient knowledge of appropriate problem-solving skills to manage in their new environment [45]. Where one resides after SCI rehabilitation is not always certain [12].

A major consequence of SCI can be the inability to return to their previous residence [46]. Some people with SCI must transition to an assisted-living facility, such as a nursing home, due to not being able to return to their prior home because of inaccessibility or not having sufficient resources and social supports [36,47].

Quality of life was determined to be lower for people with SCI living in a nursing home compared to others living in the community [46,47]. Duggan et al. [47] discovered that the social and attitudinal environment of the nursing home had the most influence on reported quality of life. Individuals of all age groups said the main reason for poor socialization was because they felt they had nothing in common with other nursing home residents, so they stayed in their room. Other negative aspects of living in a nursing home or other institutions consisted of crowded living spaces, strict schedules, lack of freedom, and unsatisfactory personal assistance [48].

Comparatively, it is more advantageous for people with SCI to live within the community, avoid social withdrawal [34] and re-engage in their life. Community reintegration and social participation have been found to improve quality of life [49], foster a sense of self-worth, confidence, competence, and an overall improvement in life satisfaction for people with SCI [50]. Such effects have been shown to have lasting positive impacts on both psychological and economic well-being [50].

Along with the person with SCI, caregivers, spouses and loved ones are instrumental regarding the feasibility of living in the community after SCI [32], along with facilitators and barriers. Facilitators can include availability of professional services (e.g., physiotherapy; personal attendants; transportation services; respite care); positive attitudes (e.g., feeling independent and appreciative); social support (e.g., structural and functional support; informational and peer support); and when the caregiver feels confident and competent. Barriers to integrating into the community can include lack of knowledge or availability of community resources, fragmented continuity of care, role strains, negative attitudes and poor coping [32], and being socially disadvantaged [30].

Environmental obstacles to participation in social events and family gatherings can be psychologically wearing. Whether there are too many steps and no elevator, or lack of privacy for personal care needs, a person’s integration into the community [26] can be vastly restricted. These constraints can trigger reminders about all of their losses and evoke a grief response [26].

Regardless of residential status, strong social support by caregivers and friends can be a major factor for successful community reintegration [32] and is especially psychologically protective. Social supports can serve to buffer inhibitions a person with SCI may have with community reintegration and bolster coping with negative experiences related to environmental obstacles [34] as well as micro and macro stigma of disability from larger society [50].

During and after rehabilitation, therapeutic interventions based on hope theory have been helpful after SCI. Hope theory involves two cognitive components: pathway thinking and agency thinking [51]. Pathway thinking refers to a person’s belief in their capacity to create “routes” to accomplish their goals based on one’s personal values, along with alternate plans for flexibility. Agency thinking refers to one’s confidence to act and accomplish the desired goal.

Both existential needs and physical needs are important for social reintegration and daily participation in greater society. Peer-based programs [32] are beneficial at all stages from acute rehabilitation to long-term care to facilitate community participation [28].

### Adaptive Technology (AT)

Adaptive technologies (ATs) have been found to, at times, both enhance and detract from people’s quality of life [52]. ATs are very individualized in complicated ways; people’s use and non-use can vary across time, situation, and person. In general, ATs are intended to increase independent functioning and promote independence and to, ideally, live in the community. In order to achieve this, it is important that AT devices are specifically matched to the individual’s unique abilities and preferences [52].

It is important to be mindful about the compatibility or incompatibility of the person with an AT, as personal characteristics and psychosocial variables have been identified as predictors for using AT [53]. In addition, problems and frustrations can be additive, and result in premature avoidance or abandonment (non-use). Therefore, the matching a person with technology (MPT) model is used to help reduce variability in use of ATs. There are three MPT areas of consideration when determining if AT may be beneficial for the person: (1) understanding how the device is used within the individual’s environmental and psychosocial settings (and if the AT will be for the acute stage of SCI, later, or continuously); (2) how the individual’s personality and temperament is related to technology use; and, (3) the notable characteristics of the AT itself [52,53]. Although these are positive guidelines to utilize, it is important to remember that MPT is a complex process as an individual’s expectations and reactions to technology devices are complex [53].

## 8. Mood and Coping

SCI and associated challenges with physical discomfort and impairments, societal inconveniences and disappointments, perceived stigma, financial strain, and perceived limitations on autonomy impart immense and demanding challenges on one’s coping resources. Despite the vast complexities of living with SCI, many people have better than projected outcomes [1] and do not experience significant problems. However, some people do experience significant problems with mood and coping following SCI.

Depression, anxiety, substance use disorders, and PTSD are significantly more prevalent for people with SCI compared to the general population. Despite the frequent interactions with health professionals, these are often undiagnosed and treated [54]. Reasons for this omission could be that the physical impairments are more salient, or problems distinguishing normal emotional reactions from pathological reactions, the presumption that extreme psychological reactions are normal, or stigma around mental health problems. In general, mental health problems prior to a disabling injury are often risk factors for mental health conditions after injury [9].

### 8.1. Depression

Prevalence of major depressive disorder among people with SCI is influenced by many factors, such as quality of social relationships [36], financial strain [37], predisposing psychological status [9,12], severity of secondary conditions [34,35], cognitive impairment [55], and subjective perceptions of control, self-esteem, and coping [56,57]. Previous studies have found that major depressive disorder occurs within 16% and 38% of the adult SCI population during rehabilitation, as well as after discharge from the hospital [56]. When left untreated, depressive symptoms remained significantly elevated for up to 2 years post-injury compared to those who receive Cognitive Behavioral Therapy (CBT) treatment [58], or symptoms can last more than a decade with no treatment [35]. Furthermore, negative mood states increase the risk of longer hospitalizations, additional medical complications, decreased independence, increased time in bed, and difficulties with transportation [58].

Noted predictors of depressed mood among people with chronic SCI were pain, poor self-efficacy, negative coping appraisals, declining health, alcohol misuse [27], and ineffective coping [35]. Depressive symptoms can be exasperated by the fact that SCI serves as a consistent reminder of loss, especially as individuals with SCI learn to navigate their new roles, their environment, health issues [12], and facing life milestones differently than before [34] or differently than expected. As a result, people living with SCI endure living losses, which require constant adaptation and adjustment [34].

Longer time since initial injury has been found to be a protective factor against depression by some research [28], yet other research suggests that passing time alone does not resolve depression [35]. Importantly, most studies have found that only about one-third of people with SCI are clinically depressed, while the majority (about 70%) are substantially resilient and stable [35,56]. Of note, our language when assessing mood may be important. Some people with SCI may not think of mood issues in a clinical sense and instead reference their “low points” as a response to life, and not the criterion from a conventional medical model [34]. Instead, they talk about general mood and responses and reactions to daily events in their life to assess mood.

### 8.2. Anxiety

Some anxiety is normal after SCI, but when anxiety becomes severe or interferes with daily functioning there is a clinical concern. About 30–45% of people with SCI will experience significant levels of anxiety [58,59]. These feelings of anxiety can be explained by the traumatic nature of the injury, fear of secondary conditions (i.e., autonomic dysreflexia), psychological predispositions, or persistent thoughts regarding their well-being and future [59,60]. These anxiety reactions seem to be unique in the sense that they occur in response to situational contexts, as opposed to be enduring in nature [61]. Nevertheless, it is important to note that some anxiety self-report measures may inflate anxiety prevalence rates due to the somatic symptom questions that overlap with SCI secondary conditions (i.e., temperature dysregulation, blood pressure, respiratory functioning, and motor weakness) [59]. Anxiety levels seem to heighten during the initial period of injury but often decline around the 1-year post-injury mark [61].

#### Longitudinal Analysis

A longitudinal analysis of the emotional impact and coping after SCI found little changes in anxiety and depression over the course of 10 years, and that the majority of people with SCI do not become depressed [35]. Coping was also found to be stable. Coping strategies utilized at week 12 were predictive of clinical conditions ten years later. Specifically, coping with positive reinterpretation was associated with less or no depression, and coping with behavioral disengagement was associated with a high probability of depression. Collectively, the findings from this study negate the common expectation that depression is an expected or inevitable result after SCI.

Importantly, neither level of injury nor severity of functional impairments was associated with amount of distress experienced, which suggests that other factors contribute more to psychological distress than the more obvious physical ramifications from the SCI [35].

### 8.3. Substance Use Disorder

Substance use disorders (SUD) are common in the general population and even more prevalent in an SCI population. For perspective, one in five Americans have a mental illness, and one in 12 have an SUD [54]. SUDs are known to cause extra suffering and impairments to people after SCI [13]. Therefore, SUDs should be assessed during SCI acute treatment, and include a lifetime history of use and abuse with alcohol, marijuana, illicit drugs, tobacco, and misuse of prescription medications. Follow-up screenings for SUDs should occur during subsequent visits.

### 8.4. Posttraumatic Stress Disorder

A vulnerability to posttraumatic stress disorder (PTSD) after SCI depends upon a combination of individual and contextual factors before (pre), during (peri) and after (post) SCI. A systematic review [62] identified factors present *before* SCI or *at the time* of SCI that may make one vulnerable to PTSD: psychiatric history, family instability, and perceived negative social supports. Factors lending to vulnerability to PTSD *after* SCI were depressed mood, negative appraisals, distress, anxiety, and pain severity. Psychosocial components such as lower education and income were also associated having PTSD following SCI.

Diagnosing PTSD in an SCI population has challenges. PTSD after critical care often goes unnoticed [63]. Survivors of SCI following intensive care hospitalization can experience ICU-related PTSD, which is clinically profiled with behaviors of avoidance, worries about re-experiencing, and produces poor engagement in rehabilitation and medical treatment. ICU-related PTSD causes symptoms that are separate from the event causing SCI and may require exploration if suspected.

Another thing to keep in mind is the symptom overlap with PTSD and depression that requires extra scrutiny when assessing people with SCI. For example, arousal, sleep problems, avoidance, and loss of interest in “usual activities” can be characteristic of either diagnosis.

## 9. Mood and Coping Summary

Rehabilitation provides an excellent opportunity to normalize psychological treatment and provide psychological interventions and education. Depression, Anxiety, SUDs, and PTSD are treatable conditions, and providers should assess for these during acute rehabilitation, and also during subsequent visits for persistence of symptoms or changes, including the development of new problems.

Responses to SCI vary greatly, as do coping strategies and appropriate coping interventions. Maladaptive coping strategies of behavioral disengagement or withdrawal, substance misuse, and denial are associated with depression, anxiety and PTSD. Adaptive coping strategies of acceptance [12], positive growth from adversity, active coping and problem-solving [35], avoidance of self-pity [34] and social support are associated with generalized positive adjustment post-SCI. Some people benefit from coping interventions that balance loss and acceptance [12]. Others found that coping interventions focusing on grief and losses were less helpful, however. Instead, learning and researching about their injury and limitations, followed by problem-solving and reflecting on present and future opportunities were most helpful. Many individuals found that a helpful coping mechanism was a conscious focus being on strength and motivation to maximize independence [34]. If transportation presents as a barrier for formal or other interventions, telemedicine may provide a viable, socioeconomical option.

Helpful support for the person and their family following SCI are not purely clinical. Instrumental support such as housekeeping or babysitting, and emotional consolation can help to reduce feelings of guilt and burdensomeness that people with SCI report [12]. Peer support, particularly early after the injury, benefits both the person with SCI and their caregiver [32]. Quality social relationships, overall, facilitate coping, modulate distress, and reduce chronic stress that is associated with chronic illness [39]. Low mood and anxiety are complex for people with SCI, and helpful solutions often involve psychosocial factors.

## 10. Adaptive and Maladaptive Behaviors

As highlighted throughout, the majority of people adapt well after SCI, and psychopathology, immediately or even years later, is not an inevitable consequence [35,64]. Adaptive behaviors following SCI manifest in many forms. Adaptive, positive behaviors not only enrich mood, but positive behaviors beget more positive behaviors, which help abate maladaptive behaviors, and can even help mitigate suicide [54]. Examples of adaptive behaviors following SCI include engaging in pleasurable and rewarding activities, pursuing avocational or vocational interests [28], accessing and utilizing health care services [30], cultivating quality social and intimate relationships [31], fostering relationships with SCI professionals [12], and obtaining cultural or spiritual beliefs about one’s meaning and value in life [54].

For the purpose of this paper, maladaptive behaviors are actions that can potentially cause harm, and fall under the assemblage of “self-harm behaviors” and give different degrees of concern, and present with a large range in variability (i.e., ranges from treatment adherence behaviors to suicidal behaviors).

### 10.1. Nonadherence Behaviors

By definition, “self-harm behaviors” are acts that one takes that may result in harm to oneself, irrespective of the apparent purpose of the act. There is much diversity in behaviors that cause self-harm which are on a continuum from mild unease to an alarming degree that necessitates professional intervention [65].

In medical settings, self-harm behaviors can present as mismanagement of medicine, missed appointments, too much or too little sleep, levels of concerning nonadherence, social isolation, alcohol or drug misuse, “giving up” statements, and decisions to medically hasten death [66]. It is helpful to think of these behaviors as “symptoms” that can provide information about the underlying contributors [66], rather than the actions are the result of clinical “depression” or defiance. For example, self-harm behaviors may be “symptoms” of problems adapting to disability, existential issues, medical mistrust, or behaviors that are simply patient preferences [67,68,69,70]. Attaining the root cause of the symptom and issue (s) underlying the behavior allows for more deliberate responses and targeted interventions.

Clinical responses should focus on the core conflict of whatever triggers the maladaptive behavior [66,68]. For example, a person may display anger, or indifference, or may inconsistently adhere to essential medical recommendations, causing others sincere worry about consequential harm. Reasons for the angry or indifferent behaviors may be problems adapting to disability, or core conflicts involving loss of autonomy, or limited coping or having an external locus of control [68]. Interventions aimed at the conflict driving the maladaptive behavior will provide the most benefit, and often best when combined with psychopharmacological treatment. Additional clinical responses in the present example surrounding adapting to disability or loss of autonomy or external locus of control could be: to promote mastery of new and existing life-valued activities, bolster coping skills, facilitate peer support, or help with meaning making [66]. All behaviors suspected of causing substantial self-harm should be assessed professionally so appropriate interventions can be put in place.

### 10.2. Suicide

Self-harm behaviors that are exceedingly worrisome can be symptoms of suicidality [71]. Behavioral symptoms of suicide risk are when a person states an intention for dying; displays warning signs like changes in behaviors, thoughts, or emotions; increases substance use; displays increased verbal or physical aggression; or shows disengagement from social contacts. Worrisome expressions are statements about preferring to be dead or desiring peace or a sense of control over their life. Additionally, emotional changes showing shame, hopelessness, guilt, anger, sadness, anxiety, or irritability should be evaluated. Selling or giving away belongings, purchasing firearms, or hoarding pills could be preparatory behaviors. All of these behaviors could be signs of suicidality and indications that a person needs immediate professional assessment and prompt interventions (https://www.mentalhealth.va.gov/suicide_prevention/index.asp (accessed on 26 February 2022).

It is of utmost importance to attend and delve further when a person with SCI says they feel like a burden to others. Joiner et al. [38] found perceived burdensomeness (usually on a loved one) as an significant identifiable feature of people who completed suicide compared to noncompleters. Burdensomeness was highly associated with using more lethal means of suicide, even when controlling other factors. Explanations for the connection between burdensomeness and suicide range from evolutionary-psychological theories to altruistically motivated suicide.

Proper assessment of the risk for self-harm is especially important in the SCI population [16]. During a moment of frustration, a person with SCI may say that they wish they were dead. This could be a statement meant to fully express the intensity of their sentiment at that moment, or they may have a genuine interest in dying. It is important to ask directly about these statements. Either under-estimating self-harm (e.g., minimize symptoms by inaccurately assuming normalcy or resilience) or over-estimating (e.g., excessive protections above actual risk) can cause harm. Furthermore, unsuitable responses can damage the patient–provider relationship, and possibly result in alienation, distrust, and an increased apathy about having an interest in living [16].

Research identified biopsychosocial risks for suicide in SCI population. Medical (biological) risk factors can be worsening of functional limitations, new major illness, and chronic pain [72]. Psychological risk factors include suicidal thoughts, prior suicidal attempts, mood or substance abuse disorders, hopelessness, personality disorders, and prior psychiatric hospitalizations [64,72]. Social risk factors for suicide can be loss of a relationship, legal problems, financial problems, and lack of functional social support. It is important to educate and provide resources on suicide risk and treatment options to family, loved ones, and caregivers.

Clinical practice Guidelines suggest using brief, evidence-based screening tools to assess suicidal ideation (thoughts) in individuals with SCI: (a) during the initial hospitalization, (b) and repeat screen to measure persistence of or change in symptoms, (c) immediately after discharge, and (d) annually, at minimum, or more often if risk factors are present [54]. Risk factors decrease as protective factors increase; therefore, it is important to assess both.

Immediate responses should include reducing access to lethal means and creating a safety plan. Removing or locking firearms, restricting medication access or individually wrapping pills can give distance between suicidal thoughts and access. A safety plan lists personalized strategies for internal coping, people to call or activities to serve as a distraction, or if these are insufficient, a list of professionals and agencies to contact for help.

## 11. Traumatic Brain Injury

Prospective studies show that individuals with SCI have a high rate of comorbid traumatic brain injury (TBI) with an estimated range of 47–74% of cases [73]. Although problems with cognition, independent functioning, and general mental health are associated with dual diagnosis of SCI and TBI [74], the presence of TBI is not a primary focus during acute rehabilitation. Instead, treatment focuses on the obvious serious spinal trauma [74]. Although there may be reasons for this diversion, individuals with concomitant TBI often do not receive proper assessment and treatment for TBI, and this leaves the true incidence and outcomes of concurrent SCI and TBI uncertain [75].

Rather than focusing on SCI and TBI as simultaneous injuries, a study with a large sample of veterans with SCI concentrated on lifetime history of TBI, and the cumulative burden on injury severity and number of traumas [55]. Over 75% of individuals with SCI were determined to have at least one brain injury in their lifetime, and almost 50% of the sample indicated a history of moderate to severe brain injury. The number of brain injuries had meaningful implications. Regardless of severity of brain injury, having more than one prior brain injury affected functional outcomes for people with SCI. In this sample, individuals with two or more brain injuries had lower health-related quality of life, and reduced functional independence in activities such as self-care, problem solving, communication, social interaction, and memory. On a positive note, history of TBI in this sample did not impact the level of independent living nor affect the person’s social relationships or impact their physical or subjective well-being.

In conclusion, research shows that individuals with a mild TBI or no TBI have the best functional and psychosocial outcomes regarding mood, health-related quality of life, and overall life satisfaction [55,76]. This supports the brain’s resilience and capacity to heal following a single, minor brain insult [76].

## 12. Sexual Health

As the result of previous literature on sexuality and persons with SCI being limited to the gender binary of male and female, this section overview will also be limited to the gender binary. However, we acknowledge that gender is a social construct that lies along a spectrum. Thus, we also recognize that more research must be focused on sexuality and gender non-binary persons with SCI.

The myth of bodily perfection and the myth of asexuality are two misconceptions born from a larger societal focus on the function of genitalia, phallocentric pleasure, and attractiveness of perfect bodies [77]. Persons with SCI may be deemed unattractive and non-sexual as a result of their injury and impairment, and hence be perceived as “different” from the societal norms and standards of body image and sexual activity [78].

Despite these misconceptions, sexual health is a significant component of SCI rehabilitation, and sexual rehabilitation is a contributing factor to self-esteem and quality of life for individuals with SCI [7,79]. In fact, sexuality as a domain in life satisfaction was rated lowest amongst other life domains [28]. Persons with SCI face many abrupt changes to their pre-injury sexual life that may cause avoidance of sexual experiences due to pain, weakness, sensory loss, and neurogenic bladder and bowel. This lack of sexual activity may cause the individual with SCI to feel lost, lonely, and helpless even if they are in a current relationship [79].

### 12.1. Men, SCI, and Sexual Health

Following a spinal-cord injury, men may experience some sexual dysfunction or dissatisfaction depending on the level of and time since initial injury along with psychosocial factors [80,81]. Some common sexual impairments for men with SCI include decreased libido, erectile and/or ejaculatory dysfunction, semen abnormalities, and anorgasmia [80]. Psychosocial factors impacting sexual response and functioning are post-SCI self-esteem and body-image, relationship status, prior sexual attitudes and experiences, and openness to sexual rehabilitation adaptation [80]. Body image issues regarding loss of muscle mass, larger stomach, pressure sores, spasticity, bladder and bowel issues, placement in a wheelchair, and flaccid penis are factors that play a particularly significant role for men’s self-perception of physical attractiveness, especially as compared to their pre-SCI bodies [81]. Body image issues may result in men feeling hopeless, having an enhanced fear of rejection, and engaging in avoidance behaviors, which often leads to increased negative self-thoughts [81].

After SCI, men can learn to become more appreciative of the psychological and emotional aspects of sex while moving away from their previous conceptualization of sex as a physical (specifically penetration) and self-serving (own pleasure) act [81]. This redefining of sex may help men with SCI find a new sense of satisfaction from their sexuality and sexual experiences post-SCI [81]. Studies show that men who prioritize aspects of intimacy (i.e., touching and caressing) and emotional connectedness with less emphasis on physical penetration were found to have a deeper and better sexual experience and level of intimacy [81,82]. Additionally, men also found a greater sense of sexual self-satisfaction from placing an importance on sexually satisfying their partner [81].

### 12.2. Women, SCI, and Sexual Health

The literature surrounding women with SCI and sexuality is far more limited due to the smaller percentage of women acquiring SCI as compared to men [83]. Similar to men with SCI, women with SCI may experience some sexual dysfunction that consists of decreased or lost sensation and anorgasmia [84]. Women with SCI may also experience body image issues, decreased self-confidence, and difficulty meeting new partners [84]. In order to experience more pleasurable sexual experiences and compensate for lack of physical sensation in the vagina and clitoris, women with SCI often need alternative methods of arousal [84]. These alternative methods may include caressing, kissing, touching of other erogenous zones with sensation, as well as experiencing feelings of romance, intimacy, consideration, and acceptance from their partner [84].

Most women with SCI resume their normal menstrual cycle approximately one year after injury, and some undergo pregnancy and giving birth [85]. When pregnant, women with SCI and their providers should be cautious of increased risk of complications such as exacerbated spasticity, autonomic dysreflexia and urinary tract infections [83].

Additionally, reseating needs of pregnant women with SCI must be considered in order to manage increased risk of respiratory issues, decubitus ulcers, and thrombosis [83]. Women with SCI and their providers should discuss labor expectations, for signs and symptoms of labor may look different or may not be experienced at all depending on the level of injury [83]. Furthermore, preparations are necessary for delivery options and outcomes due to the increased rate of complications during delivery [85] and higher incidence of preterm delivery, low birth rate, and possible need for a neonatal intensive unit [83].

Additionally, postpartum depression is comparatively more common for women with SCI and is associated with higher levels of stress, more social isolation and less satisfaction with their social network, reduced mobility, unemployment, abuse, and poorer overall health [83].

More attention and quality research is warranted in the area of women and SCI. Published surveys have indicated that the majority of women with SCI felt that they were inadequately informed about pregnancy during their childbearing years, and during pregnancy. At the same time, the majority of surveyed obstetricians admitted to being somewhat uneasy caring for pregnant women with SCI [83].

### 12.3. Sexual Health Considerations

Sexual health after SCI is a non-linear process that will change and evolve over time for both men and women with SCI. It is important to encourage exploration and experimentation of new sexual acts and experiences that may not have been practiced or tried previously. By constructing more inclusive and broader definitions of sexuality, societal myths of asexuality and unattractiveness for people with SCI may be debunked.

It is equally important to include partners of persons with SCI in sexual rehabilitation so both can learn to navigate new sexual changes together [79]. Communication is key to redefining what sex and intimacy means to both the person with SCI and their partner. Furthermore, for partners of people with SCI, it is important to remain mindful of maintaining a balance between identities of being a romantic and/or sexual partner and a caregiver in the relationship. It may be beneficial to seek out peer-counseling and support about sexual health from other couples living with SCI.

Biopsychosocial factors related to sexual health should be included during rehabilitation in order to properly recognize the physical, psychological, and emotional aspects of sexuality in the context of that person’s area for sexual concern and priorities when addressing sexual health. Specifically for women, topics concerning contraception, fertility, and pregnancy options and associated risks should be discussed with a PCP. Additionally, postpartum depression should be assessed so that appropriate referrals can be made, and interventions can be implemented. Providers should always leave the door open regarding interventions for sexual health for persons with SCI in order to mitigate feelings of intimidation or hesitancy to reach out for education and resources about sexual health.

### 12.4. Loss of and Redefining Masculinity

In addition to a sudden change in sexual health, male-identified individuals with SCI may experience a loss of the masculinity [34], as defined by social norms and traditions. These societal and traditional definitions of masculinity are commonly characterized by physical strength, assertiveness, dominance, and sexual prowess and conquests [80,81]. Additionally, masculinity is often defined by one’s possession, size, and use of their genitals [81]. Through these definitions, it is easy to see that these aspects of masculine identity affect not only sexual health, but also shapes a person’s functioning in daily life [81]. For those who strongly identify with this meaning of masculinity and sexual salience, feelings of depression, loss, insecurity, and lower self-worth may be exacerbated [80]. Furthermore, those who previously viewed independence and self-reliance as major components of their masculinity may also have difficulty engaging in emotional and social support post-injury, and thus may show increased feelings of depression as a result of their negative perception of their self with SCI [80].

In order to foster acceptance and redefine the traditional conceptualization of masculinity, men with SCI are encouraged to reflect on their own personhood and nonphysical values as they create a new meaning and understanding of their own individual masculinity [81]. For example, it may be beneficial to use an existential approach to challenge the traditional masculine concept of strength by physical build and ability, and redefine it through other means such as, strength in vulnerability, strength of character, strength in accepting support and assistance when needed, strength in processing and expressing one’s true emotions, etc. [81]. Additionally, normalizing and educating patients about other ways to experience sexual pleasure that do not require men to act in the dominant role during sexual interactions may alleviate negative feelings about masculinity [81].

### 12.5. Loss of and Redefining Femininity

Just as male-identified individuals with SCI may face loss of masculinity, female-identified individuals with SCI may also experience a loss of femininity, although there is much less literature about this topic for women [77]. Of the existing literature on the topic of femininity among individuals with SCI, women report that the biggest loss is not feeling physically attractive through the lens of both their own self and others [77]. Women with SCI also report feeling concern or a sense of failure as a result of believing they can no longer provide sexual fulfillment to their partner based on their own negative perception of physical attractiveness post-injury [77].

Female-identified individuals with SCI have femininity obstacles that differ from the non-SCI female experience. For instance, others may damagingly perceive women with SCI as “twice as feminine” in the sense that they are viewed as weaker and more passive as a result of their female identity *and* disability [86]. Furthermore, femininity may also be conceptualized as it relates to specific gender roles and accomplishments (i.e., ideal body image, domestic tasks, childbearing, motherhood, etc.) [86]. Therefore, women with SCI may have an added pressure to achieve these standards in order to “prove” their femininity [86].

It is important to encourage female-identified individuals with SCI to reflect on their conceptualization of femininity after their injury and redefine it as they see fit. This redefinition of femininity may expand to brainstorming creative ways in which women with SCI can express their femininity with their current SCI appearance (i.e., focusing on hair, make-up, jewelry, etc.) [86]. Furthermore, it may be necessary to normalize and educate women with SCI that motherhood/childbearing and femininity are not mutually exclusive, and to explore pregnancy options and alternatives if desired [86].

## 13. Aging with SCI

The age and stage of life a person acquires SCI will fundamentally affect the totality of psychosocial consequences, and how disability is integrated into one’s life path [1,2,12]. The amount of time one lives with an SCI is momentous to the amount of change one experiences over the course of one’s life, and affects one’s existential interpretation of their SCI [2]. A consistent finding, independent of the age one acquires SCI or the number of years living with SCI, is that life satisfaction remains a vital component [28], and adaptation and change are continuous regardless of time since injury [31].

Over time, some things become more challenging and some things easier for people with SCI, or for people with disabilities in general [2,26]. Aging with SCI presents additional challenges to natural aging such as more than average number of secondary impairments [2,26]. These vulnerabilities and other complications from SCI can become cumulative and impede one’s ability to stay resilient [1]. Conversely, there were noted areas where people with SCI and other disabilities reported life became easier. Largely, as people with disabilities age, they felt less need to prove themselves. Compared to their younger selves, people aging with disability expressed having a healthier acceptance of themselves, undergoing forgiveness of themselves regarding parenting, and withstanding the pressure to exert more energy than they have [2]. In addition, research shows that older people with SCI were less concerned with their physical appearance [19,28].

Retirement generated diverse reactions for people with SCI [2,40]. Some happily retired early, declaring no need to conform to age expectations and feeling content [2]. Some depicted retirement as placing them on a level playing field with others. Retiring at age 65 promoted an enhanced sense of belonging, and feeling like they were finally equal with peers in one of life’s elements. On the other hand, others become morose after retirement. Leaving one’s job also meant leaving coworkers, often a constant form of social interaction [40] that was removed with retirement. Self-stigma can also increase after retirement. In addition to social connectedness, being in the workforce helps shield felt stigma and societal discrimination [2].

## 14. Post-Traumatic Growth

Psychopathology is not inevitable following SCI [58,61]; most people demonstrate resilience and procure good outcomes despite multiple biopsychosocial challenges [1,24,31,61]. In fact, stress-related growth, benefit finding, and posttraumatic growth (PTG) occur for many people following SCI [87], and PTG can also happen for caregivers, or people caring for someone with SCI [25]. PTG may be a new term, but the concept that great good can come from great suffering is planted in many religions, literature, and myths. By definition, PTG refers to a qualitative change in a person, for the better, after trauma; PTG is more transformative than hardiness, optimism, sense of coherence, or resilience that one may utilize to manage and cope during periods of extreme distress [88].

A large study found that most people experienced some positive change after SCI, with the greatest change being a sense of personal fortitude and realizing that they were stronger than previously thought [34,87]. A study by Kalpakijian [87] found that being younger and female were associated with PTG. Authors conjected that this finding may relate to how women process and consider different dimensions of an experience, and how younger individuals may be more open to change. Interestingly, the level or severity of SCI was not related to PTG, and PTG after SCI was similar to PTG after other traumatic events.

Tedeschi and Calhoun’s [88] well-cited model states that PTG results from the struggles a person has while discovering a new reality following a trauma, not necessarily the trauma itself. In quick summation, the person pre-trauma encounters a significant disruption of their assumptive worlds, resulting in stress and a breakdown of personal narratives and beliefs and goals. This is then followed by deliberate ruminations and a schema change, and subsequently evolves into a newly developed narrative and enhanced wisdom. How one governs the struggle will determine the extent of growth and wisdom achieved. Studies by Tedeschi and colleagues found that PTG most often prevails in personalities that are open to experience and extraversion, people possessing cognitive processing styles that permit schema changes, and a social environment conducive for reconstructing personal narratives [88].

Distress, at times, is a normal experience, and people with PTG are not exempt from emotional distress [87,88]. What appears most important to retain and further propagate PTG was to approach distress in a manner of acceptance, and address challenges with active coping [61]. In addition, having a sense of strong social supports and quality relationships helps people retain resilience that can foster PTG [1]. Lastly, volunteering, peer counseling or sharing one’s experience after SCI can enhance one’s personal growth [27].

## 15. Conclusions

Humans are complex and every person with SCI is extremely unique. As shown in this review, SCI profoundly affects multiple dimensions of a person’s psychological and social well-being, with an assortment of consequences and responses ranging from remarkably adaptive to gravely worrisome. Ranges in severity level of injury, functional impairments, quality social supports, experienced and perceived stigma, financial and vocational strain, predisposing and/or secondary mental and physical health conditions, and sexual health concerns are all individual factors that continually interact with one another as resulting consequences of SCI.

Providers in health care systems are the first contacts a person with a newly acquired SCI will interact with; therefore, there is a certain accountability to mindfully consider “the person” and associated psychosocial elements amidst the complex medical care.

Figure 2 summarizes issues presented in this paper. This illustrates how the person with SCI is the center and key factor to all clinical outcomes. The outside circles show the mediating influences, with loved ones and caregivers having a more direct effect on the experiences for the person with SCI. The relationships often negotiate or buffer the several described psychosocial consequences. The psychosocial consequences are overlapping entities, and each can shape the next. This all occurs in the context of attitudes of others, access, community, and larger society. A few individuals report having a transformative experience after SCI despite the challenging psychosocial consequences after SCI, called post-traumatic growth.

This review, by no means, covers all possible psychosocial consequences of SCI. Humans are vastly unique and complex and our aim was to offer some of the most foundational challenges and changes many individuals and their loved ones may experience after spinal cord injury. This review is also limited with acknowledging distinctive psychosocial consequences for congenital spinal cord injuries and pediatric spinal cord injuries. Medical consequences of SCI are thoroughly addressed in other papers in this Special Edition.

Breaking away from ableist attitudes, bias, and stigma of persons with disability is vital to lessening the psychosocial aftermath from an SCI. Reconstructing a positive view of living with disabilities and realizing that most people do adapt and effectively manage all associated challenges can facilitate meaning making and life satisfaction for those with disabilities. Hopefully, then, the consequences after SCI will be less challenging.

## Figures and Tables

**Figure 1 jpm-12-01178-f001:**
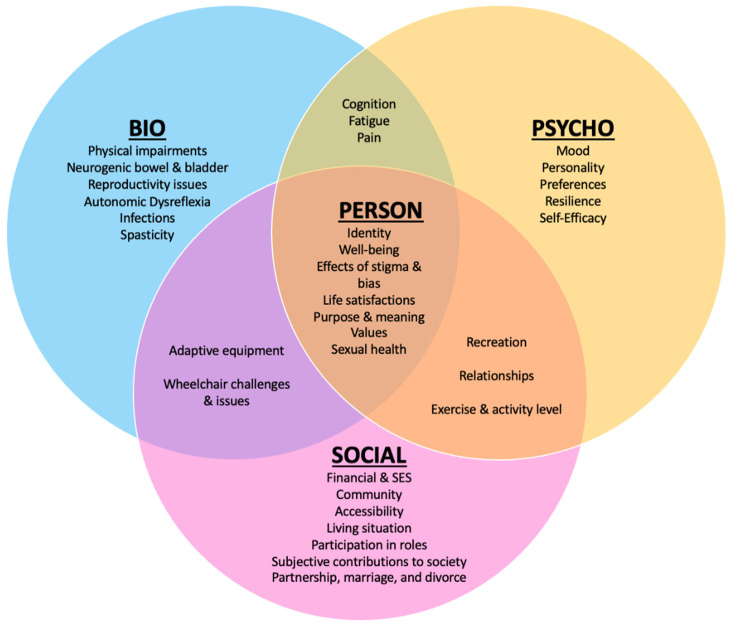
Biopsychosocial model of interacting factors for persons with SCI.

**Figure 2 jpm-12-01178-f002:**
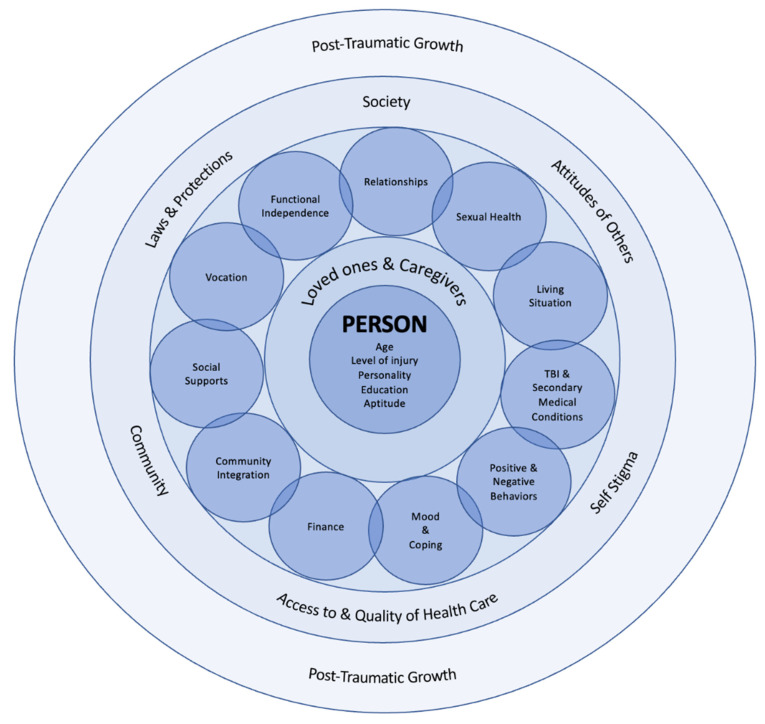
Model illustrating some of the psychosocial consequences a person experiences after acquiring SCI.

## Data Availability

Not applicable.

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
