# Peer review of "Psychosocial Consequences of Spinal Cord Injury: A Narrative Review"

_jpm, 2022, doi:10.3390/jpm12071178_

Round 1
Reviewer 1 Report
Major Concerns
- The section ‘Things to consider’ is a little confusing. On one hand, it seems this is intended for clinicians to understand what to take away from the sections. This entire review is a big ‘things to consider’. Basically, you are aggregating previous studies in the field, and discussing it. So what exactly are we to make of this ‘things to consider’ section?. On the other hand, it looks like a glorified conclusion of each section. Some paragraphs at the end of each section summarizing the core points expressed in the section will be better.
I am guessing the authors are writing this review for a very much larger audience. Listing suicide prevention telephone numbers etc, in the ‘things to consider’ section feels like a clinical brochure or some form of clinical manual for everyone, and not a review paper for scientists and clinicians. It seems the authors have a much wider audience or a rather smaller one in SCI?. I am very much confused here.
If this is critically important, please consider explaining the rationale for having a ‘things to consider’ section (Line 63). And do not include a ‘things to consider’ already in the intro (Lines 120-125).
- The aim of this review paper is not so clearly stated. In the last paragraph of the introduction, please consider clearly stating the aim of this review. This should preferably be stated before the ‘psychosocial consequences’ paragraph (see minor concerns #2 below). …..
- In Lines 765-766 you wrote ‘As shown in this manuscript, SCI profoundly affects multiple dimensions of a person’s biological, psychological, and social well-being, with an assortment of 766 responses from remarkably adaptive to gravely worrisome’. Please consider a modification to this sentence as you did not discuss the biological implications of SCI….you clearly focused on the psychosocial.
Minor Concerns
- Please consider replacing the word manuscript with either ‘paper’ or ‘review’ where necessary. Eg. ‘……this paper will attempt to parse out and review psychosocial…’ Line 55
- Too many subtitles in each section does not make for a good read. In the introduction ‘APPRECIATING THE ABLEIST WORLD IN WHICH PEOPLE WITH SCI LIVE’, there are two subsections A, and B. Consider making them paragraphs, and not subsections. This makes one intro section. After which aim of review should be clearly stated (see point number 2 of Major concerns above).
- Please consider adding a table of contents to the beginning of the paper, to structure it well.
The figure 2, summarizes very well what the manuscript is about. It should be brought up a bit to the psychosocial consequences section.
Author Response
We would like to thank the reviewers for their constructive feedbacks and comments which have improved the readability of this paper. We would like to make the Editor aware that we have followed the suggestions sent by the reviewers. We have made significant edits in the paper using the track changes mode in MS Word throughout the text. We will include a point-by-point response to the comments:
REVIEWER #1:
Major Concerns:
- “The section ‘Things to Consider’ is a little confusing… On one hand, it seems this is intended for clinicians to understand what to take away from the sections…”
Response: After reading your comments, we agree and removed all “Things to Consider” subsections. Much of the content from these was deleted and the elements fitting for a scholarly review were embedded into the larger text, or provided as a summary per your suggestion.
“I am guessing the authors are writing this review for a very much larger audience…. Or a smaller one in SCI? I am very much confused here.”
Response: You are correct. The task of this article is to present the literature on the “psychosocial consequences of SCI,” for the readers of a special edition on Spinal Cord Injury in JPM. This topic is very broad and, as you stated -- has much “to consider” (pun intended).
- “The aim of this review paper is not so clearly stated. In the last paragraph of the introduction, please consider clearly stating the aim of this review. This should preferably be stated before the ‘psychosocial consequences’ paragraph.”
Response: You are correct, again. We added a paragraph describing the purpose and aims of the paper. We provided an overview of the paper here, too.
- “In Lines 765-766 you wrote, ‘As shown in this manuscript, SCI profoundly affects multiple dimensions of a person’s biological, psychological, and social well-being, with an assortment of responses from remarkably adaptive to gravely worrisome.’ Please consider a modification to this sentence as you did not discuss the biological implications of SCI.. you clearly focused on the psychosocial.”
Response: We modified the sentence to clearly state the review exclusively focused on psychosocial elements.
Minor Concerns:
- “Please consider replacing the word manuscript with either ‘paper’ or ‘review’ where necessary.”
Response: We made the suggested changes.
- “Too many subtitles in each section does not make for a good read. In the introduction… Consider making them paragraphs, and not subsections.”
Response: We agree and substantially reduced the number of subtitles.
- Please consider adding a table of contents to the beginning of the paper to structure it well.
Response: This is a helpful suggestion. However, this would deviate from JPM’s conventional presentation. We added material to help the reader know the outline and pre-identify the order of the material.
- The Figure 2 summarizes very well what the manuscript is about. It should be brought up a bit to the psychosocial consequences section.
Response: Authors discussed the suggestion of moving Figure 2 to the beginning as an overview. However, we prefer to keep it at the end as a summary of the holistic review of psychosocial consequences after acquiring a SCI. We altered the introduction to present a better “overview” of the paper.
Reviewer 2 Report
Dear Authors, unfortunately I find the manuscript full of methodological gaps and too many narrative bypasses of the most common drafting conventions.
Abstract should not provide potential description of the development of the manuscript, but a background, objective, results and discussion and finally a conclusion
L43 In reality, these concepts have been embraced by the much deeper classification model of the ICF
The manuscript is structured in a purely informative way, I would suggest reshaping it as a scoping review. It looks like a chapter in a treatise on cognitive psychology.
There is no objective outlined. The overview is wide, but far too digressive. There are eyelets with few recent bibliographic references.
The limitations are totally missing. It doesn't look like a scientific manuscript.
Author Response
We would like to thank the reviewers for their constructive feedbacks and comments which have improved the readability of this paper. We would like to make the Editor aware that we have followed the suggestions sent by the reviewers. We have made significant edits in the paper using the track changes mode in MS Word throughout the text. We will include a point-by-point response to the comments:
REVIEWER #2:
- “Dear authors, unfortunately I find the manuscript full of methodological gaps and too many narrative bypasses of the most common drafting conventions.”
Response. Thank you for agreeing to read and review the paper. The purpose of the paper is to present an overview of the multiple psychosocial consequences following SCI and not a formal review or research project. We made substantial revisions in the paper that improves the scholarly content while containing practical information for practicing clinicians or readers who are interested in responding to the psychosocial consequences following a spinal cord injury.
“Abstract should not provide potential description of the development of the manuscript, but a background, objective, results and discussion and finally a conclusion…”
Response. Thank you for the constructive feedback. We modified our abstract within the context of the type of this paper.
“L43 In reality, these concepts have been embraced by the much deeper classification model of the ICF. The manuscript is structured in a purely informative way… There is no objective outline…”
Response. Thank you for directing our attention with these comments. We added a paragraph outlining the content of the article.
“The limitations are totally missing….”
Response. Thank you, we added statements about the limitations in the conclusions.
Reviewer 3 Report
The authors have suggested a review for psychosocial consequences following SCI. Overall, the manuscript is very well written and organized. The following subtitles were outlined:
1. Relationships/family
2. Finances and employment
3. living situation
4. community integration
5. mood/coping
6. self harm
7. TBI
8. Sexual health
9. Aging
10. Post traumatic growth.
Were these sub-categories identified from a comprehensive survey from the SCI community or the authors articulating based on literature review? I really feel the list is comprehensive and I don't have much to add. One concept that I was thinking about as I read the entire paper was how do these changes compare to non-SCI population? Or individuals experiencing other medical complications (i.e. cancer, amputations, etc.)
I will provide a few comments where questions/considerations were discovered:
1. Relationships and role changes -Line 185-187 on page 5: "relationships change after SCI...May describe friendships gradually "fade away" after SCI..." I am really curious on how this differs from non-SCI (or disabled population). In general, I feel like as people age, friendships and relationships change understanding that lifestyles change naturally.
Also, how does technology impact psychosocial aspects? This could impact relationships but also coping, community integrations, etc.
2. Vocation - the authors emphasize the return to work and its positive impact on life satisfaction. Any guidelines on when or how quickly individuals need to progress to this level of integration?
5. Mood and coping - line 400-401 page 10. "However, some people do experience significant problems with mood and coping following SCI". Who are these people? can we predict early on who will have more coping difficulties and try to mitigate concerns early on?
5. 1 Depression - line 418 - CBT -??? please write out.
5.4. Substance Use Disorder - line 464-465 "Substance use disorders are common in the general population and even more prevalent in an SCI population." Do the authors have data to support this? Or does reference #12 have numbers to report?
5.5 Post traumatic Stress Disorder - line 530 page 14. "Instrumental support such as housekeeping or babysitting, and emotional consolation goes far with helping to..." - please choose a different descriptor besides "goes far". Maybe reword this sentence.
6.1 nonadherence behaviors - overall this section lacks citations. Maybe this is an area that isn't studies often, however, many bold statements are presented here and should have citations attached. Line 577 on page 15 of this same section: please add the word "can" to "All behaviors suspected of causing substantial self-harm...interventions can be put in place".
6.2 Suicide - line 582 page 15. A citation is warranted after the first sentence of this paragraph also.
8.0 Sexual Health - line 663-665 of page 17. Why is the first paragraph in italics? I don't see other sections with intalics.
9.0 Aging with SCI - again, citations needed at line 834 and 843
Figures 1 and 2 are very comprehensive. In the version provided for review, the words are very small and hard to read.
Overall, I wish the authors would provide some recommendations for health providers to break away from the albeist attitudes, biasness, and stigma provoking treatments to maximize these categories for those living with SCI, however, I realize that is likely another manuscript.
Author Response
Please see the attachment.
The authors have suggested a review for psychosocial consequences following SCI. Overall, the manuscript is very well written and organized. The following subtitles were outlined:
Thank you for taking the time and your helpful feedback.
- Relationships/family
- Finances and employment
- living situation
- community integration
- mood/coping
- self harm
- TBI
- Sexual health
- Aging
- Post traumatic growth.
Were these sub-categories identified from a comprehensive survey from the SCI community or the authors articulating based on literature review? I really feel the list is comprehensive and I don't have much to add. One concept that I was thinking about as I read the entire paper was how do these changes compare to non-SCI population? Or individuals experiencing other medical complications (i.e. cancer, amputations, etc.)
These sub-categories were identified based on a literature review. This is a great question you raise. SCI tends to differ from other disability populations as people who acquire SCI are often younger in age, are most often cognitively intact post-injury, and have years of life left to live post-injury (thus having to adapt to aging with SCI in addition to adapting to the initial injury itself, and also normal issues with advancing age, as noted in the section on aging and SCI).
I will provide a few comments where questions/considerations were discovered:
Relationships and role changes -Line 185-187 on page 5: "relationships change after SCI...May describe friendships gradually "fade away" after SCI..." I am really curious on how this differs from non-SCI (or disabled population). In general, I feel like as people age, friendships and relationships change understanding that lifestyles change naturally.
We have addressed this with a sentence on lines 167-168.
Also, how does technology impact psychosocial aspects? This could impact relationships but also coping, community integrations, etc.
This is a great questions and important aspect of many people’s lives who are living with SCI. We decided to include a sub-section titled “Adaptive Technology” in order to address the impact of technology on the psychosocial aspects of an individual.
- Vocation - the authors emphasize the return to work and its positive impact on life satisfaction. Any guidelines on when or how quickly individuals need to progress to this level of integration?
Unfortunately, there are no one-size-fits-all guidelines on when or how quickly individuals need to progress to this level of vocational integration. Newer models emphasize vocational reintegration depends on the person’s pace according to their individual needs and goals.
- Mood and coping - line 400-401 page 10. "However, some people do experience significant problems with mood and coping following SCI". Who are these people? can we predict early on who will have more coping difficulties and try to mitigate concerns early on?
Prevalence of mood disorders among people with SCI is influenced by many factors, such as quality of social relationships, financial strain, predisposing psychological status, severity of secondary condition, cognitive impairment, and subjective perceptions of control, self-esteem, and coping. Additionally, people who already struggle with mood and coping problems pre-SCI are more likely to be at risk for continued problems post-injury.
Regarding early prediction about future coping after SCI, a longitudinal analysis by Pollard & Kennedy (2007) found coping strategies utilized at week 12 were predictive of clinical conditions ten years later. These facts and details are in the paper.
- 1 Depression - line 418 - CBT -??? please write out.
The following edit has been made in order to clarify, in text: Cognitive Behavioral Therapy (CBT).
5.4. Substance Use Disorder - line 464-465 "Substance use disorders are common in the general population and even more prevalent in an SCI population." Do the authors have data to support this? Or does reference #12 have numbers to report?
The reference cited in this sentence has data that supports this claim.
5.5 Post traumatic Stress Disorder - line 530 page 14. "Instrumental support such as housekeeping or babysitting, and emotional consolation goes far with helping to..." - please choose a different descriptor besides "goes far". Maybe reword this sentence.
This sentence has been reworded as you suggested, thank you.
6.1 nonadherence behaviors - overall this section lacks citations. Maybe this is an area that isn't studies often, however, many bold statements are presented here and should have citations attached.
We appreciate this observation and have revised to include multiple new references throughout this sub-section in order to support the statements made, thank you.
Line 577 on page 15 of this same section: please add the word "can" to "All behaviors suspected of causing substantial self-harm...interventions can be put in place".
This has been inserted into the sentence.
6.2 Suicide - line 582 page 15. A citation is warranted after the first sentence of this paragraph also.
We have included a citation from Bostwick (2015) Ref #71 in order to support this first sentence.
8.0 Sexual Health - line 663-665 of page 17. Why is the first paragraph in italics? I don't see other sections with intalics.
This is stylistically italicized as an acknowledgement that previous research on this subject is limited to gender-binary demographic samples, and that sufficient research on this topic as it relates to people with SCI who identify as gender non-binary is still needed in the field.
9.0 Aging with SCI - again, citations needed at line 834 and 843
We have updated both sentences with the appropriate citations.
Figures 1 and 2 are very comprehensive. In the version provided for review, the words are very small and hard to read.
Thank you. We have enlarged the figured in order to hopefully make the text easier to read.
Overall, I wish the authors would provide some recommendations for health providers to break away from the albeist attitudes, biasness, and stigma provoking treatments to maximize these categories for those living with SCI, however, I realize that is likely another manuscript.
We are happy to hear that you recognize these types of recommendations are extremely important if health providers are to break away from any kind of ableist attitudes, biases, and stigma regarding care for people with SCI. You are right to assume that this is likely another manuscript in itself! This topic is actually a passion of the authors and we are planning to bring it to life through its own deserved manuscript with a pilot project being currently reviewed.

Round 2
Reviewer 1 Report
The authors aswered all my concerns.
Author Response
Thank you very much for your contributions that improved our paper.
Reviewer 2 Report
Dear authors,
I had already suggested my decision regarding the manuscript, unfortunately I do not consider the revision of the paper satisfactory. I had suggested an edit through the ICF classification, for a more scientific facet. Furthermore, the paper is considered as a qualitative study, but then it is drawn up as a scoping review, without however respecting its writing traits.
Author Response
Thank you for taking the time to respond to the edited version of the paper. We appreciate the ICF model and do not feel there is a discrepancy with how we framed the paper using the biopsychosocial model of disability, which is a “model of disability” that fits within the ICF classification. We felt the best way to organize the material was using the biopsychosocial model since our focus was on “psychosocial consequences” following a spinal cord injury, a population with disabilities, and this model is how the ICF is described as a model of disability. Please refer to page two of this document, “The ICF: An Overview” from the CDC and WHO. The ICF model embraces multi-dimensional concepts that “is a biopsychosocial model of disability” that integrates the social and medical models of disability. We noted earlier, our task was to exclusively focus on psychosocial consequences as other papers will speak to the medical elements of SCI. Under the figure on page two of the CDC/WHO working page, states, “Although personal factors are recognised in the interactive model shown in Figure 1, they are not classified in the ICF at this time. Such factors influence how disability is experienced by the individual and some, such as age and gender, are commonly included in data collections.” We hope that this explanation is satisfactory regarding your suggestion. We are working with the JPM editors regarding the appropriate classification of this paper. Thanks again for promoting and endorsing science in our field.
PT6 Working paper (cdc.gov)
This manuscript is a resubmission of an earlier submission. The following is a list of the peer review reports and author responses from that submission.